# Spatial distribution and determinants of HIV high burden in the Southern African sub-region

**Olatunji O. Adetokunboh**[1,2]*, **Elisha B. Are**[1,3]

**1** DSI-NRF Centre of Excellence in Epidemiological Modelling and Analysis (SACEMA), Stellenbosch University, Stellenbosch, South Africa, **2** Department of Global Health, Division of Epidemiology and Biostatistics, Stellenbosch University, Cape Town, South Africa, **3** Department of Mathematics, Simon Fraser University, Burnaby, BC, Canada

\* olatunji@sun.ac.za

## Abstract

### Background

Spatial analysis at different levels can help understand spatial variation of human immuno-deficiency virus (HIV) infection, disease drivers, and targeted interventions. Combining spatial analysis and the evaluation of the determinants of the HIV burden in Southern African countries is essential for a better understanding of the disease dynamics in high-burden settings.

### Methods

The study countries were selected based on the availability of demographic and health surveys (DHS) and corresponding geographic coordinates. We used multivariable regression to evaluate the determinants of HIV burden and assessed the presence and nature of HIV spatial autocorrelation in six Southern African countries.

### Results

The overall prevalence of HIV for each country varied between 11.3% in Zambia and 22.4% in South Africa. The HIV prevalence rate was higher among female respondents in all six countries. There were reductions in prevalence estimates in most countries yearly from 2011 to 2020. The hotspot cluster findings show that the major cities in each country are the key sites of high HIV burden. Compared with female respondents, the odds of being HIV positive were lesser among the male respondents. The probability of HIV infection was higher among those who had sexually transmitted infections (STI) in the last 12 months, divorced and widowed individuals, and women aged 25 years and older.

### Conclusions

Our research findings show that analysis of survey data could provide reasonable estimates of the wide-ranging spatial structure of the HIV epidemic in Southern African countries. Key determinants such as individuals who are divorced, middle-aged women, and people who recently treated STIs, should be the focus of HIV prevention and control interventions. The

**Data Availability Statement:** All data files are available from the figshare database (URL https://figshare.com/s/33e95ee4594a7c146e3b).

**Funding:** The study was funded by the Sub-committee B, Stellenbosch University. This work

was also supported by the South African Centre for Epidemiological Modelling and Analysis (SACEMA), which receives core funding from the Department of Science & Innovation, Government of South Africa the National Research Foundation (NRF). The funders had no role in study design, data collection and analysis, decision to publish, or preparation of the manuscript

**Competing interests:** The authors have declared that no competing interests exist.

spatial distribution of high-burden areas for HIV in the selected countries was more pronounced in the major cities. Interventions should also be focused on locations identified as hotspot clusters.

# Background

## Introduction

Human immunodeficiency virus (HIV) infection and acquired immunodeficiency syndrome (AIDS) have claimed more than 30 million lives over the last three decades [1]. At the end of 2020, the total number of people living with HIV/AIDS was estimated to be about 37.7 million globally, with the majority residing in sub-Saharan Africa [2].

To end the HIV/AIDS epidemic, the Joint United Nations Programme on HIV/AIDS (UNAIDS) proposed the 90-90-90 strategy [3]. Achieving the 90-90-90 targets in combination with several preventive strategies is expected to reduce HIV incidence, prevalence, and AIDS-related mortality by almost 80% in the next ten years [3]. By the end of 2021, about 85% of the people living with HIV knew their status, 75% were on antiretroviral treatment (ART) and only 68% were virally suppressed [2].

The progress in addressing the HIV epidemic is dynamic, and many sub-Saharan African countries have achieved notable advancements in reducing new infections and improving access to treatment and care; however, several Southern African countries are still battling relatively high HIV prevalence rates. This makes the area a region of interest that needs to be thoroughly investigated [2].

## Spatial patterns of HIV risk

Spatiotemporal analysis at national and sub-national levels can help understand spatial variation of HIV infection, disease drivers, and targeted interventions; however, this type of study is limited in sub-Saharan Africa [4]. Prior studies on the mapping of HIV epidemics and those at higher risk of infection have contributed to understanding the characteristics of HIV hotspots and spatial heterogeneity [5]. Unfortunately, the non-availability of good quality spatially referenced data for most African countries hinders regular and detailed spatial studies of HIV prevalence [5]. Nevertheless, spatial distribution and patterns play a vital role in the transmission of HIV in sub-Saharan African countries [6].

A geospatial analytical study characterizing areas with high levels of HIV transmission among seven countries in Eastern and Southern Africa regions shows that, among young adults, there were areas with relatively high prevalence alternating with low prevalence areas, signifying the presence of areas with high levels of HIV transmission [7]. However, this study is limited by the restriction of the participants to only young adults (15–29 years of age) and data that were collected as far back as 2008. Using data obtained from all adult age groups is expected to be more robust and informative than focusing on a single age group, as there are important differences in epidemiology, biology, and behavior across age groups [8]. A study on mapping HIV prevalence in sub-Saharan Africa shows significant local variations in the rate and direction of change in HIV prevalence between 2000 and 2017 in the region [9]. These findings demonstrated how significant local variations could be masked when looking at trends at the national level.

### Determinants of HIV infection

Prior studies have shown that some socioeconomic and behavioral factors can significantly influence the likeliness of acquiring HIV infection. For example, Bärnighausen et al. show that acquiring an additional year of education in a poor South African rural community decreases the risk of developing HIV by up to 7%, when other key factors such as sex, age, wealth, place of residence, partnership status, are controlled for [10]. Similarly, in Lesotho, being formally educated is negatively associated with HIV infection. Also, married women who had extra-marital affairs were less likely to use a condom than non-married women [11]. However, these studies and several others were either restricted to just a locality or conducted more than a decade ago. Therefore, updated research that uses recent data across Southern African countries is necessary.

In this study, we evaluated the determinants of the HIV burden in Southern African countries. The study also explored the spatial distributions of high HIV prevalence in Southern African countries.

## Methods

### Site selection

The focus of this study was on the Southern African sub-region and involved six countries: Malawi, Mozambique, Namibia, South Africa, Zambia, and Zimbabwe. These countries were selected based on the availability of demographic and health surveys (DHS), between 2011 and 2020. They were also selected based on their high HIV prevalence and availability of the corresponding geographic coordinates of sampled locations.

### Data collection

The survey data for this study was a cross-sectional representative sample of households extracted from the Measure DHS website (http://www.dhs.program.com) [12]. The sampling frame for the surveys consisted of enumeration areas across each included country. The surveys consisted of a two-stage sample design with standardized questionnaires administered to the participants [12]. Trained interviewers gave questionnaires to selected participants in each of the included countries during the data collection stage. Permission was granted for data extraction by Measure DHS and the data were downloaded in STATA format.

DHS survey samples were selected using a stratified, two-stage cluster design with enumeration areas as the primary sampling unit and the households as the secondary sampling units in each of the participating countries. The DHS program was funded by various development partners, including the Global Fund to Fight AIDS, Tuberculosis and Malaria, the United States Agency for International Development, etc., and technical assistance from MEASURE DHS, ICF International, Calverton, Maryland, USA. The corresponding National Ministry of Health or Research/Statistical agencies implemented the DHS program.

### HIV data

For the HIV data used, eligible women within the age range of 15–49 years and men aged 15–59 were encouraged, during the interview, to test for HIV voluntarily. Trained interviewers afterward collected finger-prick dried blood spot specimens for HIV counseling and testing. The data collection and analysis details were based on anonymously linked protocols. The DHS program methodology allows for combining HIV test results with other information gathered based on each respondent's unique questionnaire.

The outcome was defined as an individual that tested positive for HIV infection during the survey. Determinant variables included in this study are as follows: participant's gender, the age of the participants in completed years, educational status, employment status, wealth index, marital status, and place of residence. Other included variables are the number of sexual partners in a lifetime, sexually transmitted infections (STI) in the last 12 months, HIV testing and the regular use of condoms during sex with the most recent partner in the previous 12 months [13,14].

### Geospatial data

The geospatial data of the respondents included the latitude and longitude coordinates of their residence to allow the mapping of individual and community data. In addition, the study constructed a geolocated database of HIV prevalence data from surveys.

### Data analysis

This study is a secondary data analysis from country-wide and community-based surveys. HIV prevalence for each included country was analyzed using the most recent DHS data. The analyses processes were as follows:

a. The logistic regression analyses of associations between HIV infection and independent variables were undertaken with results reported as odds ratios (ORs) with their corresponding 95% confidence intervals (CIs). Crude (unadjusted) odds ratio results stem from a straightforward model incorporating only one variable at a time. In contrast, adjusted odds ratios (AOR) have been modified to accommodate additional predictor variables in the model.

b. The relationship between HIV status and the determinant factors was evaluated using the chi-square test for independence.

c. HIV datasets were merged to Global Positioning System (GPS) coordinates using GADM database of Global Administrative Areas v 4.1.[15].

d. The study assessed the presence and nature of HIV spatial autocorrelation for each country, using the Global Moran's *I* statistic. Significant clusters are geographic areas in which HIV prevalence is higher than in neighboring areas. The presence of spatial autocorrelation is suggestive of HIV clustering and is indicative of hierarchical expansionary spread across districts. Moran's *I* positive values show the presence of spatial autocorrelation; negative values denote divergent values clustered next to one another while zero means absolute spatial randomness. Moran's *I* (p < 0.05)—statistically significant result indicates the presence of spatial autocorrelation. The analysis was done using STATA 16.0 [16].

e. To produce smooth surfaces of HIV prevalence for data generation at the subnational level, we used the Heatmap (Kernel Density Estimation) spatial interpolation method for each selected country and data years. The spatial analysis was conducted using (QGIS Development Team [2020]. QGIS Geographic Information System. Open-Source Geospatial Foundation Project. http://qgis.osgeo.org").

f. Data cleaning and recording were done with STATA 16.0.

## Results

### Study population

Six Southern African countries, namely Malawi, Mozambique, Namibia, South Africa, Zambia, and Zimbabwe were included in this study. Each of the countries and the number of respondents is shown in Table 1. The overall prevalence of HIV based on the DHS varied between

**Table 1. Study population and frequency distribution of HIV cases among selected Southern African countries.**

| Variable | Survey year | Sample | Prevalence | HIV positivity | |
| --- | --- | --- | --- | --- | --- |
| | | | | Female* | Male* |
| Malawi | 2016 | 14 779 | 1803 (12.2) | 1 066 (13.5) | 737 (10.8) |
| Mozambique | 2015 | 11 270 | 1661 (14.7) | 1 181 (17.3) | 480 (10.8) |
| Namibia | 2013 | 8 858 | 1233 (13.9) | 814 (16.3) | 419 (10.8) |
| South Africa | 2016 | 4 862 | 1089 (22.4) | 773 (28.4) | 316 (14.8) |
| Zambia | 2018 | 24 702 | 2802 (11.3) | 1 814 (13.8) | 988 (8.6) |
| Zimbabwe | 2015 | 16 475 | 2472 (15.0) | 1 583 (17.5) | 889 (12.0) |

* Number of participants tested positive for HIV and percentage in parenthesis.

11.3% in Zambia (2018) and 22.4% in South Africa (2016). The HIV prevalence rate was higher among female respondents in all six countries.

## Relationship between determinants of socio-demographic factors and HIV prevalence

Table 2 shows the relationship between socio-demographic factors determinants and the prevalence of HIV infection among respondents in the selected countries. About half of the respondents had secondary or higher education, two-fifths were within the 15–24 years age group, and one-third were never married. The table also shows significant associations between HIV status and independent determinants such as age, marital status, employment status, wealth index, lifetime sex partners, gender, residence, regular use of condoms with recent partners, HIV testing and recent STI. The overall HIV prevalence in the six countries surveyed was 13.7%. Respondents aged 35 years and older, those ever married, female, those living in rural settings, having three or more lifetime sex partners and those who have ever tested for HIV had higher HIV prevalence (Table 2).

## Summary result of logistic regression analyses

The crude and adjusted odds ratios by the socio-demographic and behavioral characteristics of the respondents for the six Southern African countries were summarized in (Tables 1–6 in S1 File). Across the countries, being older, divorced or widowed, and those who had STIs in the last 12 months had significantly higher odds of being HIV positive. Respondents who are divorced, separated, or widowed had significantly higher odds of being HIV positive than those who had never been married among the countries (Malawi AOR 2.57, 95% CI 1.89–3.48; Mozambique AOR 2.88, 95% CI 2.23–3.72; South Africa AOR 1.61, 95% CI 1.15–2.23; Zambia AOR 3.00, 95% CI 2.46–3.67; Zimbabwe AOR 3.62, 95% CI 2.83–4.64). The odds of HIV positive status were higher in older respondents compared to 15–24 years old respondents (Malawi AOR 3.11, 95% CI 2.60–3.73; Mozambique AOR 2.33, 95% CI 1.95–2.79; Namibia AOR 4.96, 95% CI 3.82–6.44; South Africa AOR 5.01, 95% CI 3.87–6.50; Zambia AOR 4.66, 95% CI 3.95–5.50; Zimbabwe AOR 4.20, 95% CI 3.52–5.02).

Among the male respondents, the odds of being HIV positive were lesser than that of female respondents (Mozambique AOR 0.79, 95% CI 0.68–0.90; Namibia AOR 0.66, 95% CI 0.56–0.77; South Africa AOR 0.43, 95% CI 0.36–0.51; Zambia AOR 0.56, 95% CI 0.50–0.62; Zimbabwe AOR 0.72, 95% CI 0.64–0.81). The probability of HIV infection was higher among those who had STIs in the last 12 months than those who did not report any STI (Malawi AOR 2.00, 95% CI 1.51–2.65; Mozambique AOR 1.94, 95% CI 1.53–2.46; Namibia AOR 1.77, 95%

**Table 2. Results of HIV positivity and association between determinants and HIV infection among six Southern African countries.**

| Variables | Sample size | Percentage (%) | HIV infection | | Percentage HIV positive (%) | p-value |
|---|---|---|---|---|---|---|
| | | | No* | Yes* | | |
| HIV positive | 11 060 | 13.7 | | | | |
| HIV negative | 69 886 | 86.3 | | | | |
| *Age* | | | | | | |
| 15–24 years | 32 581 | 40.3 | 30 851 (38.1) | 1 730 (2.2) | 5.3 | <0.001 |
| 25–34 years | 22 303 | 27.5 | 18 708 (23.1) | 3 595 (4.4) | 16.1 | |
| 35+ years | 26 062 | 32.2 | 20 327 (25.1) | 5 735 (7.1) | 22 | |
| *Education* | | | | | | |
| No formal education | 6 101 | 7.6 | 5 233 (6.5) | 868 (1.1) | 14.2 | 0.225 |
| Primary | 32 237 | 39.8 | 27 796 (34.3) | 4 441 (5.5) | 13.8 | |
| Secondary/Higher | 42 608 | 52.6 | 36 857 (45.5) | 5 751 (7.1) | 13.5 | |
| *Marital status* | | | | | | |
| Never married | 29 683 | 36.7 | 27 419 (33.9) | 2 264 (2.8) | 7.6 | <0.001 |
| Ever married | 51 263 | 63.3 | 42 467 (52.5) | 8 796 (10.9) | 17.2 | |
| *Wealth index* | | | | | | |
| Poor | 26 982 | 33.4 | 23 608 (29.2) | 3 374 (4.2) | 12.5 | <0.001 |
| Middle | 26 983 | 33.3 | 23 256 (28.7) | 3 727 (4.6) | 13.8 | |
| High | 26 981 | 33.3 | 23 022 (28.4) | 3 959 (4.9) | 14.7 | |
| *Employment status* | | | | | | |
| None | 45 461 | 56.2 | 38 789 (47.9) | 6 672 (8.3) | 14.7 | <0.001 |
| Employed | 35 485 | 43.8 | 31 097 (38.4) | 4 388 (5.4) | 12.4 | |
| *Gender* | | | | | | |
| Male | 36 268 | 44.8 | 32 439 (40.1) | 3 829 (4.7) | 10.6 | <0.001 |
| Female | 44 678 | 55.2 | 37 447 (46.3) | 7 231 (8.9) | 16.2 | |
| *Residence* | | | | | | |
| Rural | 31 301 | 38.7 | 26 120 (32.3) | 5 181 (6.4) | 16.6 | <0.001 |
| Urban | 49 645 | 61.3 | 43 766 (54.1) | 5 879 (7.3) | 11.8 | |
| *Number of lifetime sex partners* | | | | | | |
| 1 | 16 899 | 28.9 | 15 524 (26.5) | 1 375 (2.4) | 8.1 | <0.001 |
| 2 | 13 005 | 22.3 | 10 956 (18.7) | 2 049 (3.5) | 15.8 | |
| 3+ | 28 603 | 48.8 | 23 370 (39.9) | 5 233 (8.9) | 18.3 | |
| *Sexually transmitted infections in the last 12 months* | | | | | | |
| No | 69 735 | 86.3 | 68 022 (84.2) | 1 713 (2.1) | 2.5 | <0.001 |
| Yes | 11 033 | 13.7 | 10 394 (12.9) | 639 (0.8) | 5.8 | |
| *Ever tested for HIV* | | | | | | |
| No | 20 187 | 25.0 | 18 919 (23.4) | 1 268 (1.6) | 6.3 | <0.001 |
| Yes | 60 735 | 75.0 | 50 945 (62.9) | 9 790 (12.1) | 16.1 | |
| *Used condom every time had sex with a most recent partner in last 12 months* | | | | | | |
| No | 3,840 | 25.5 | 3,087 (20.5) | 753 (5.0) | 19.6 | 0.013 |
| Yes | 11,232 | 74.5 | 8,816 (58.5) | 2,416 (16.0) | 21.5 | |

CI 1.28–2.46; South Africa AOR 1.98, 95% CI 1.44–2.72; Zambia AOR 2.42, 95% CI 1.96–2.99; Zimbabwe AOR 2.52, 95% CI 1.92–3.30). The odds of HIV positive status were higher in people who ever tested for HIV compared to those who had never tested (Mozambique AOR 1.70, 95% CI 1.47–1.97; Namibia AOR 2.57, 95% CI 1.98–3.34; South Africa AOR 1.38, 95% CI 1.06–1.79; Zambia AOR 1.49, 95% CI 1.22–1.83; Zimbabwe AOR 1.49, 95% CI 1.24–1.77).

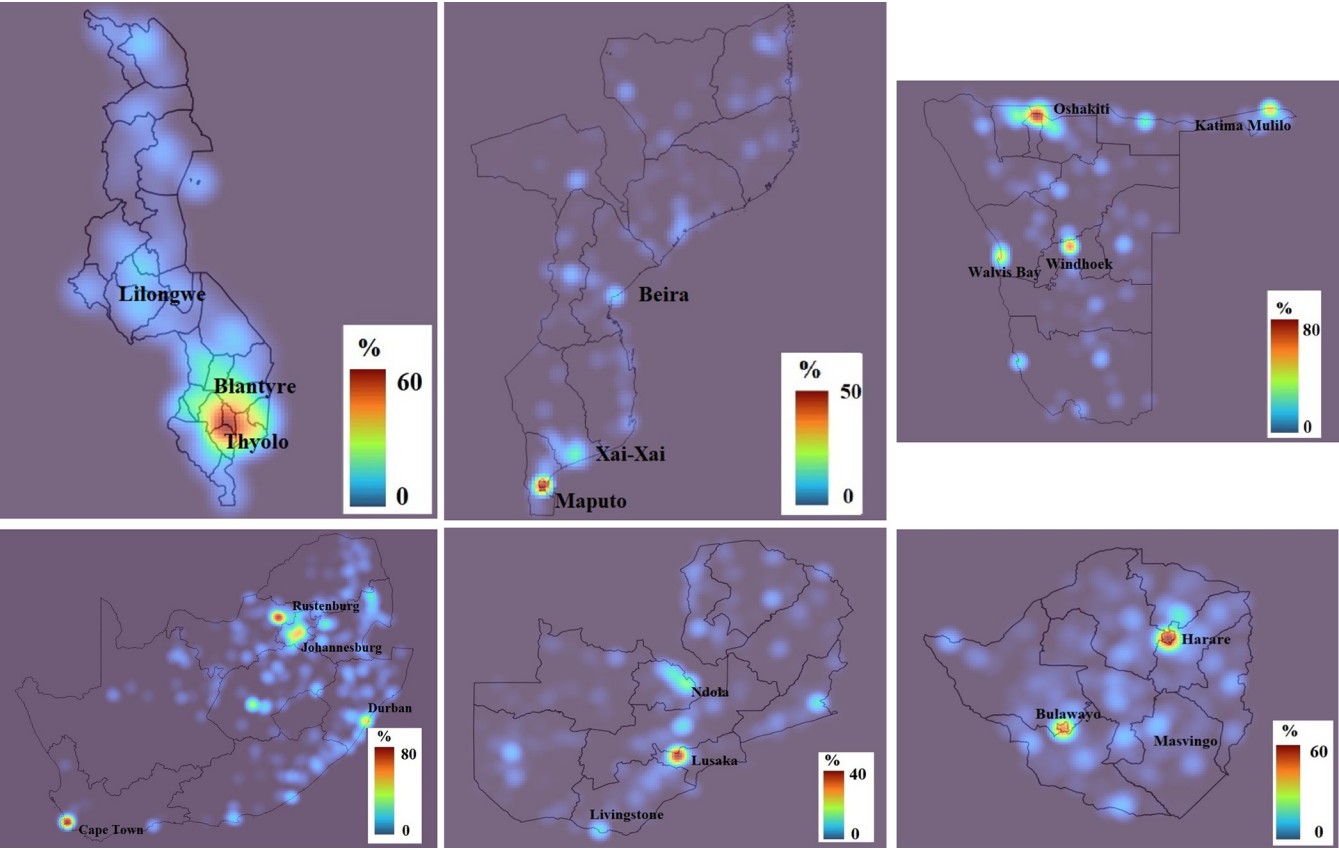

**Fig 1. A.** A heatmap showing the HIV prevalence in Malawi estimated by kernel estimator approach (2016). **B.** A heatmap showing the HIV prevalence in Mozambique estimated by kernel estimator approach (2015). **C.** A heatmap showing the HIV prevalence in Namibia estimated by kernel estimator approach (2013). **D.** A heatmap showing the HIV prevalence in South Africa estimated by kernel estimator approach (2016). **E.** A heatmap showing the HIV prevalence in Zambia estimated by kernel estimator approach (2018). **F.** A heatmap showing the HIV prevalence in Zimbabwe estimated by kernel estimator approach (2015).

**Spatial patterns of HIV.** This study shows the areas within the hotspot cluster windows for each included country as seen in Fig 1A–1F. Communities within and around Oshakati and Windhoek in Namibia; Maputo in Mozambique; Blantyre in Malawi; Cape Town, Durban, Rustenburg, and Johannesburg in South Africa; Ndola and Lusaka in Zambia; and Bulawayo and Harare in Zimbabwe are key identified high clusters of HIV cases.

The presence of statistically significant clusters at various significant levels for different surveys was also identified as seen in the (Figs 1–6 in S2 File). The study shows a Low-High HIV occurrence and cluster around the Central Region of Malawi; this implies that the area has low HIV prevalence, but its neighboring clusters have high HIV prevalence; High-Low cluster in Zambezia Province—the area has a high HIV prevalence, but the adjacent areas have low values for the same variable; and the High-High cluster in Maputo Province of Mozambique—the area has a high HIV prevalence, likewise the neighboring clusters. In addition, there were also Low-Low clusters in the Omaheke, Khomas and Hardap Regions of Namibia–the clusters and neighboring clusters have low HIV prevalence; High-High cluster in Southern Province of Zambia; and Low-High cluster in Bulawayo Province and High-Low cluster in Mashonaland East Province of Zimbabwe. There was no significant clustering among the South African Provinces. The positive Moran's *I* value shown in Table 3 reveals that neighboring provinces/

Table 3. HIV prevalence Moran's I among selected Southern African countries.

| Country | Moran's I | Z | p-value |
|---------|-----------|---|---------|
| Malawi | 0.158 | 8.388 | <0.001 |
| Mozambique | 0.319 | 8.897 | <0.001 |
| Namibia | 0.180 | 6.934 | <0.001 |
| South Africa | 0.150 | 5.880 | <0.001 |
| Zambia | 0.307 | 11.290 | <0.001 |
| Zimbabwe | 0.046 | 1.598 | 0.110 |

regions tend to have similar prevalence rates for HIV except in the case of Zimbabwe which has a non-statistically significant positive Moran's I value.

## Discussion

This study provided a visual and empirical analysis of spatial variation and trends in HIV prevalence among adults in selected Southern African countries from 2011 to 2020. The hotspot cluster findings show that the major cities in each country are the key sites of high HIV burden. This is a unique study that used different Southern African datasets and triangulation approaches to build logistic regression models of the determinants and spatial distribution of HIV infection.

The trends of HIV prevalence in the last decade show the magnitude of changes and the effectiveness of various HIV control activities in selected Southern African countries. While Malawi, Namibia, Mozambique, Zambia, and Zimbabwe have shown a gradual reduction in HIV prevalence over the years, South Africa at a time had an increase before declining [2]. This study also shows that South Africa has the highest HIV prevalence among the countries of study [2]. South Africa recorded the highest new HIV infections across all age groups, at 15% in 2020 [17]. The high HIV burden is likely to persist if there is a high number of new infections every year as reported in South Africa [17]. Zimbabwe and Malawi had the most pronounced HIV prevalence decline between 2011 and 2020 [2]. The reduction in the HIV burden in these countries could be attributed to reduction in sexual-risk behaviors. Manzou et al. ascribed the reduction in risky behaviors to religious practices; however, this may just be one of the contributory factors among many others [18]. HIV incidence has also been falling in several Southern African countries in recent years, possibly due to natural dynamics of the epidemic, widely available ART, several preventive activities [19], or a combination of all these factors.

Additionally, age, marital status, employment status, wealth index, gender, residence, regular use of condoms with recent partners, HIV testing and recent STI were all significant determinants of HIV prevalence in different Southern African countries. These findings are like some sub-Saharan African studies [5,20,21]. The higher level of HIV prevalence among Southern African females compared to their male counterparts has been previously reported [22–24]. Women are predisposed to higher HIV infection exposure, possibly due to their low socioeconomic status compared to males, which further increase their propensity to for involving in transactional and commercial sex work. These women are also subjected to cultural practices such as polygamy and widow inheritance that engender disproportionate HIV prevalence among women [25]. Individuals who are married, separated, divorced, or widowed, likely have a higher rate of HIV infections because they are currently or previously married and tend to have more sexual partners than never-married individuals [18]. HIV infection is assumed to be prevalent among the poor and with a lesser number among the middle and upper social class [26]. Gaumer et al. show that wealth effects on HIV prevalence are usually smaller and

statistically insignificant; however, a high wealth index was not associated with higher HIV prevalence, while low wealth was associated with a higher risk among certain sections of the population [27]. The findings of this study on the association of HIV positivity and wealth index vary among the countries. Individuals from Namibian and South African middle-class and rich households were less likely to test positive for HIV while wealthier participants from Mozambique and Zambia were more likely to test positive.

Bulstra et al. show several micro-epidemics of the high burden of HIV in localities with very low prevalence, thereby demonstrating the presence of zones with alternate levels of HIV transmission [7]. High HIV prevalence among sub-Saharan African young adults was also partially rationalized by the interaction between several factors viz a vis; economic activity, environmental, behavioral, socioeconomic, and environmental factors that were predictive of high transmission settings [7]. A spatial analytic study of factors associated with HIV infection in Malawi shows that the southern region has the highest prevalence, with the southeastern part of the country having a more pronounced HIV epidemic [28]. Nakazwe et al. show there was a decline in the overall HIV prevalence among women and men [29]. However, HIV prevalence increased among urban young men–the suggestive of disparity variance effects of HIV prevention efforts between urban and rural areas [29]. The South African HIV prevalence trend analyses of antenatal care attendees show an increase from 24.8% in 2001 to 30.7% in 2017, with Kwazulu-Natal Province having the highest prevalence [30].

All the identified hotspots are located within the major cities in selected countries. Previous studies show significant variability with hotspots clustered around big cities, main transport routes, truck stops, and large population settings [5,31–33]. Gray et al. showed that population centers in several African countries are amply interconnected with the HIV epidemic [34]. The interconnectivity is due to the increased movement of individuals across African countries because of increased globalization. Thus, geographic factors play a significant role in understanding HIV prevalence, especially in studying local heterogeneity [6]. Findings by Gibbs et al. also show that HIV prevalence is highly patterned geographically around informal urban settlements in South Africa [35]. A South African study shows that the distribution of HIV infections can be localized within certain hyperepidemic communities [36]. These communities mostly have a homogeneous population, a high percentage of single individuals, and those engaged in intergenerational sex [36]. Some of the HIV high burden localities such as Rustenburg and Ndola are mining communities that attract truck drivers, miners, adolescent girls and young women. Big cities like Johannesburg, Harare, Blantyre, and Maputo have rising urbanization and migration from neighboring high burden districts, thereby leading to high prevalence.

Estimates of HIV prevalence at the community/regional level will help to channel and efficiently use limited resources and target HIV control interventions in the HIV high burden settings of Southern Africa. The findings will support the distribution of antiretrovirals needed for the treatment of people living with HIV/AIDS (PLWHA) proportionately in line with the World Health Organization (WHO) recommendation of ART and the UNAIDS fast-track treatment and care strategy for all PLWHA [37]. Previous studies show the roles of HIV geographical hotspots as local sources of HIV transmission and their use for targeted prevention approaches which could subsequently decrease the burden of HIV [38–40]. Decision-makers can use spatial analysis findings to understand the local epidemic and generate evidence for HIV programming and resource allocation [41]. Furthermore, there is also the need to evaluate HIV estimates at the district and municipality levels. Investigating at the local level will pinpoint areas with high numbers of people living with HIV/AIDS.

We used kernel density estimation (KDE) because it can effectively detect clusters of events within a given geographic area. KDE is a hotspot mapping technique that produces a smooth and continuous surface map that illustrates gradients of the variation in event intensity across

the study fields without being constrained by theme borders [42]. Its striking visual appeal makes it a popular mapping technique and one of the frequently used approaches for point pattern analysis [42,43]. KDE differs from other mapping techniques in that it uses a weighting function based on a constant bandwidth or search radius to generate a surface based on a non-parametric estimate of the intensity function across cell grids [44]. We can obtain the bandwidth and then determine the smallest distance at which clustering is most intense and significant [42].

This work contributes significantly to the existing body of knowledge by incorporating two fundamental approaches: geospatial analysis and multivariable logistic regression. By merging spatial pattern analysis with growth rate assessments and logistic regression, we achieve a profound, more insightful, and reliable exploration of geographic, HIV, and sociodemographic data. This combined approach taps into the individual strengths of each technique, fostering a more comprehensive grasp of intricate phenomena. This method is a valuable framework in various domains, including public health, resource management, urban planning, and environmental science. The outcomes will data drive public health strategies, policy development, resource distribution, and the understanding of trends over time, all of which are crucial for combating the disease and enhancing the lives of those affected by HIV.

The main limitation of this study is aligned to the non-inclusion of some Southern African countries such as Botswana, Eswatini and Lesotho, which have high HIV burdens and were not included due to the non-availability of recent DHS data. This study is also subjected to a modifiable areal unit problem whereby different results may be obtained from analysis of the same data. HIV prevention approaches adopted across the countries at different periods warrant caution in interpreting the findings of surveys that were collected a long time ago compared to those that are more recent.

## Conclusions

Our findings highlight the significance of regional differences in HIV prevalence as well as the impact of sociodemographic determinants and sexual behaviors. Our results suggest that analysis of subnational data could provide reasonable estimates of the wide-ranging spatial structure of the HIV epidemic in selected Southern African settings. However, similar analyses should be conducted at district and municipality levels to assess community-level patterns. The spatial distribution of high burden areas for HIV in the selected countries was more pronounced in the major cities. Other measures such as prevention and identification of STIs should also be prioritized. Determinants such as individuals who are divorced or widowed, middle-aged women, and people who recently treated STIs, should be the focus of HIV prevention and control interventions in the Southern Africa Region. The findings of this study are useful in developing HIV prevention programs as it identifies the areas and communities that require additional resources and attention. Healthcare policymakers especially in resource-constrained communities in the Southern African region can apply our findings to identify areas to target for the design of interventions to reduce HIV transmission and inform the prevention and control programs. Lastly, HIV control interventions such as HIV/AIDS awareness campaigns, ART services and adherence support for those on ART should be focused on locations identified as hotspot clusters.

## Supporting information

**S1 File. Tables 1–6. Factors associated with HIV positivity identified by multivariable logistics regression.**
(DOCX)

**S2 File. Figs 1–6.** Local Spatial Autocorrelation cluster maps.
(DOCX)

## Acknowledgments

The authors gratefully thank the Demographic and Health Surveys Program for making available the datasets for this study.

## Author Contributions

**Conceptualization:** Olatunji O. Adetokunboh.

**Data curation:** Olatunji O. Adetokunboh.

**Formal analysis:** Olatunji O. Adetokunboh.

**Funding acquisition:** Olatunji O. Adetokunboh.

**Investigation:** Olatunji O. Adetokunboh.

**Methodology:** Olatunji O. Adetokunboh.

**Project administration:** Olatunji O. Adetokunboh.

**Supervision:** Olatunji O. Adetokunboh.

**Validation:** Olatunji O. Adetokunboh, Elisha B. Are.

**Visualization:** Olatunji O. Adetokunboh, Elisha B. Are.

**Writing – original draft:** Olatunji O. Adetokunboh.

**Writing – review & editing:** Elisha B. Are.

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
