## [Decision Letter · Decision Letter 0]

22 Mar 2023

PONE-D-22-32245Spatial distribution and determinants of HIV high burden in the Southern African sub-regionPLOS ONE

Dear Dr. Adetokunboh,

Thank you for submitting your manuscript to PLOS ONE. After careful consideration, we feel that it has merit but does not fully meet PLOS ONE’s publication criteria as it currently stands. Therefore, we invite you to submit a revised version of the manuscript that addresses the points raised during the review process.

We look forward to receiving your revised manuscript.

Kind regards,

Azmeraw Ambachew Kebede, MSc

Academic Editor

PLOS ONE

Journal Requirements:

2. Thank you for including your ethics statement:  "N/A".  

a. For studies reporting research involving human participants, PLOS ONE requires authors to confirm that this specific study was reviewed and approved by an institutional review board (ethics committee) before the study began. Please provide the specific name of the ethics committee/IRB that approved your study, or explain why you did not seek approval in this case.

b. Please provide additional details regarding participant consent. In the ethics statement in the Methods and online submission information, please ensure that you have specified (1) whether consent was informed and (2) what type you obtained (for instance, written or verbal, and if verbal, how it was documented and witnessed). If your study included minors, state whether you obtained consent from parents or guardians. If the need for consent was waived by the ethics committee, please include this information.

Reviewers' comments:

Reviewer's Responses to Questions

**Comments to the Author**

1. Is the manuscript technically sound, and do the data support the conclusions?

Reviewer #1: Yes

Reviewer #2: Partly

2. Has the statistical analysis been performed appropriately and rigorously? 

Reviewer #1: Yes

Reviewer #2: Yes

3. Have the authors made all data underlying the findings in their manuscript fully available?

Reviewer #1: Yes

Reviewer #2: Yes

4. Is the manuscript presented in an intelligible fashion and written in standard English?

Reviewer #1: Yes

Reviewer #2: No

5. Review Comments to the Author

Reviewer #1: Dear Editor,

Thank you for inviting me to review this interesting manuscript.

The authors are addressing HIV, an epidemic whose incidence and prevalence in Southern Africa has remained high despite various interventions in the past 4 decades. Their research aimed at further evaluating the drivers of this epidemic in Southern Africa is relevant. While national figures are relevant, they may fall short of revealing the specific geographical areas and populations with high rates of HIV as well as the targetable risk factors specific to these populations and settings. Hence this manuscript, arising from secondary data analysis with timeseries and spatial autocorrelation methods, in which the authors have attempted to bridge this gap is worthy of publication in your prestigious journal.

The abstract is well written and gives a succinct account of the research question and the main results of the study. Although brief, the literature review reveals the main risk factors of HIV in the region and the gap in knowledge that still need s to be addressed. The authors have used appropriate research methods and analysis of the data. Generally, the data is well presented in scientific tables and figures. The discussion is coherent and well written. The conclusions are appropriate and relevant study limitations were highlighted.

The target audience may receive the authors’ message with better clarity if the following issues are considered:

• The data in Table 4 (HIV prevalence univariate local Moran’s I among selected Southern African countries) is scanty. The authors should add the columns with z-values and P values.

• The country specific autocorrelation figures (S1_6 figures) look beautiful, however, only the hot spots can be discerned also with some difficulty. The cold spots are not easy to see. The geographical regions of various countries, apart from the cities, are also not labelled on the maps yet in the discussion the authors regularly refer to them. I propose that the authors easily visible maps with clearer colour distinctions according to the different HIV prevalence and appropriately label the relevant provinces/geographical locations that were compared during data analysis.

• One of the key results is that urbanization is a common feature associated with clustering of high HIV prevalence areas in all the six countries. However, there could be a few peculiar drivers of the epidemic in some urban areas that are not generalizable. A case in point is Rustenburg which is a mining town in South Africa. The authors could further strengthen the message in the manuscript by identifying and discussing the possible drivers of HIV in the peculiar hot spots in the various countries.

• There is a noticeable trend in the reduction in the prevalence of HIV all the other countries except South Africa. In the discussion, the authors have alluded to the possible interventions that worked. A brief elaboration of these would make their point clearer to the audience. Secondly, it will be quite interesting if the authors discuss how the successful interventions relate to the spatial clustering observed in the current study. After reading the nice discussion thus far, the audience is left with some questions:

o Did the various countries use different or similar strategies for the prevention of HIV? Could there be country specific factors that hinder the successful outcome of similar interventions in South Africa?

• In the discussion the authors stated that a poor wealth index was associated with the increased risk of HIV infection. The data in table 3 to an ordinary reader seems to suggest that participants in the middle and higher wealth index bracket had higher prevalence of HIV. Secondly in the country specific multivariate logistic regressions (Supplementary file S1) a poor wealth index was an independent risk factor for HIV in only Malawi, Namibia and South Africa. The authors need to further discuss this point so as to better explain their conclusion in view of the findings as indicated in the data.

• In the conclusion, the authors (supported by the data in the current study) have advocated a focus on people with recently treated STIs as one of the targets for the control of HIV. Just a thought: would enhancement of the prevention of STIs be a more feasible strategy?

• One limitation of the use of spatial autocorrelation is the phenomenon of modifiable areal unit problem (MAUP) whereby different results may be obtained from analysis of the same data, grouped into different sets of areal units. It is not clear if the geographical units for analysis in the various countries were of similar size.

Reviewer #2: In this paper, Adetokunboh and Are present an analysis of the most recent DHS data from six Southern African countries. The authors identify several predictors for HIV positive status as well as geographical hotspots for HIV within these countries.

I have the following comments:

Major:

1) This is essentially a relatively standard statistical analysis of DHS data: the key results of the DHS surveys are publicly available, and the dataset is freely available for researchers to analyse. It is surprising to me that such analysis has not already been done. I propose the authors to first run a systematic literature search to check what studies using this data have already been conducted. The main results are no surprise and in my view it is already common knowledge that factors such as high age or residing in an urban setting are associated with positive HIV status.

2) The DHS data has a great amount of different variables. How did the authors choose the variables that were included in the analysis? Most variables were such that are already known to some extent correlate with HIV status, so the added value of this study seems limited. If the authors used a systematic approach to select the variables for the analyses (or e.g. followed the results of a previous analysis), this should be described in detail.

3) It is a bit unclear why the AIDSinfo prevalence data were included in this analysis. The analysis of time trends seems completely independent of the DHS data? Please note that the AIDSinfo estimates are themselves a result of models that use different data sources, so to me a simple description of such estimates does not count as original research.

Detailed comments:

4) Abstract, Methods: Please be more specific of the actual analyses that you conducted (multivariable regression analysis of DHS data; autocorrelation analysis; etc)

5) Introduction, last sentence: 73% suppression is essentially equivalent with the achievement of the 90-90-90 target, so no need to distinguish

6) Methods: Please be more clear that you used the DHS data and describe the data according to the usual DHS structure so that it's clear what datasets were used.

7) The type of analysis seems not to be mentioned anywhere. Presumably it was logistic regression?

8) Be careful about the terms: an analysis adjusted for several variables is multivariable, not multivariate, and if you report crude odds ratios, these are results of univariable, not multivariable, analysis

9) Under data analysis point d), it is unclear what you mean by “recommended”.

10) Data analysis point e), how is the timing (“for each of the available data years”)? To my understanding only one (the latest) dataset from each country was analysed?

11) Results: The paragraph “HIV prevalence trend…” is difficult to understand. The methods for estimating the yearly decrease are not given - did you use some formal statistical method, or is this simply the difference between 2020 and 2011 divided by 10 on the geometric scale? Since the direction in South Africa changed, how is it possible to give an annual increase until 2020? As suggested in the earlier comment, the added value of this analysis is unclear as it’s not related to the DHS data

12) In my view, the most interesting results are those of the geospatial analysis. It would be nice to show these figures in the main manuscript.

13) Discussion, second paragraph: It seems that prevalence and incidence are mixed up in this paragraph (and for example, it is unclear what “15% new infections in RSA in 2020” actually means). Please note that high prevalence is not necessarily a negative thing: with the expansion of ART coverage, people living with HIV are now living much longer and thus their share of the total population should not decrease too rapidly. The fact that HIV prevalence has continued to increase in RSA over time doesn’t necessarily imply that the epidemic is not under control (although this can of course be one reason). To my understanding the study only reported prevalence, not incidence, data, so nothing certain can be said about the actual epidemic control.

14) Moreover, I strongly disagree that reduction in HIV prevalence would be a result of religious practices. During the study period, all mentioned countries have moved from the limited ART coverage and CD4 dependent thresholds etc towards universal ART access to all PLHIV, and it is well known that successful ART minimized onward transmission. I would argue that the decrease in prevalence (and in particular incidence) is mainly attributable to high ART coverage, increased knowledge, less stigma etc.

15) The language needs some (minor) editing.

6. PLOS authors have the option to publish the peer review history of their article (what does this mean?). If published, this will include your full peer review and any attached files.

Reviewer #1: No

Reviewer #2: No

---

## [Author Response · Author response to Decision Letter 0]

6 Jun 2023

Response to reviewers

Dear Editor and Reviewers,

Please find below the responses to the reviewers' comments:

Reviewer #1: Dear Editor,

1. Thank you for inviting me to review this interesting manuscript.

The authors are addressing HIV, an epidemic whose incidence and prevalence in Southern Africa has remained high despite various interventions in the past 4 decades. Their research aimed at further evaluating the drivers of this epidemic in Southern Africa is relevant. While national figures are relevant, they may fall short of revealing the specific geographical areas and populations with high rates of HIV as well as the targetable risk factors specific to these populations and settings. Hence this manuscript, arising from secondary data analysis with timeseries and spatial autocorrelation methods, in which the authors have attempted to bridge this gap is worthy of publication in your prestigious journal.

The abstract is well written and gives a succinct account of the research question and the main results of the study. Although brief, the literature review reveals the main risk factors of HIV in the region and the gap in knowledge that still need s to be addressed. The authors have used appropriate research methods and analysis of the data. Generally, the data is well presented in scientific tables and figures. The discussion is coherent and well written. The conclusions are appropriate and relevant study limitations were highlighted.

Response: Thank you for the comments.

2. The target audience may receive the authors’ message with better clarity if the following issues are considered:

• The data in Table 4 (HIV prevalence univariate local Moran’s I among selected Southern African countries) is scanty. The authors should add the columns with z-values and P values.

Response: Thank you for the advice. We have created new columns for the Z and p values.

3. • The country specific autocorrelation figures (S1_6 figures) look beautiful, however, only the hot spots can be discerned also with some difficulty. The cold spots are not easy to see. The geographical regions of various countries, apart from the cities, are also not labelled on the maps yet in the discussion the authors regularly refer to them. I propose that the authors easily visible maps with clearer colour distinctions according to the different HIV prevalence and appropriately label the relevant provinces/geographical locations that were compared during data analysis.

Response: Thank for the comments, some of the hot and cold spots has been labelled.

4. • One of the key results is that urbanization is a common feature associated with clustering of high HIV prevalence areas in all the six countries. However, there could be a few peculiar drivers of the epidemic in some urban areas that are not generalizable. A case in point is Rustenburg which is a mining town in South Africa. The authors could further strengthen the message in the manuscript by identifying and discussing the possible drivers of HIV in the peculiar hot spots in the various countries.

Response: Thank you for raising this issue and giving an example. We have identified some of the driving factors for some of the urban areas.

5. • There is a noticeable trend in the reduction in the prevalence of HIV all the other countries except South Africa. In the discussion, the authors have alluded to the possible interventions that worked. A brief elaboration of these would make their point clearer to the audience. Secondly, it will be quite interesting if the authors discuss how the successful interventions relate to the spatial clustering observed in the current study. After reading the nice discussion thus far, the audience is left with some questions:

o Did the various countries use different or similar strategies for the prevention of HIV? Could there be country specific factors that hinder the successful outcome of similar interventions in South Africa?

Response: Thank you for the comments and suggestions. There were multiple and similar interventions across the countries. They were all part of the US PEPFAR programs. South Africa prevalence trend was not of failure per see despite the increase in annual average prevalence rate. The country has a good proportion of patients on ART and viral suppression; however, the issue is the huge number of new HIV infections constantly increases or stagnates the burden on yearly basis. 

6. • In the discussion the authors stated that a poor wealth index was associated with the increased risk of HIV infection. The data in table 3 to an ordinary reader seems to suggest that participants in the middle and higher wealth index bracket had higher prevalence of HIV. Secondly in the country specific multivariate logistic regressions (Supplementary file S1) a poor wealth index was an independent risk factor for HIV in only Malawi, Namibia and South Africa. The authors need to further discuss this point so as to better explain their conclusion in view of the findings as indicated in the data.

Response: Thank you for the suggestions. We added more information on the wealth index findings. 

7. • In the conclusion, the authors (supported by the data in the current study) have advocated a focus on people with recently treated STIs as one of the targets for the control of HIV. Just a thought: would enhancement of the prevention of STIs be a more feasible strategy?

Response: Thank you for this comment. Prevention and treatment of STIs should be prioritized. 

8. • One limitation of the use of spatial autocorrelation is the phenomenon of modifiable areal unit problem (MAUP) whereby different results may be obtained from analysis of the same data, grouped into different sets of areal units. It is not clear if the geographical units for analysis in the various countries were of similar size.

Response: Different areal units were used for the countries, and this was stated as part of the limitations.

Reviewer #2: In this paper, Adetokunboh and Are present an analysis of the most recent DHS data from six Southern African countries. The authors identify several predictors for HIV positive status as well as geographical hotspots for HIV within these countries.

Response: Thank you for the comments.

I have the following comments:

Major:

1) This is essentially a relatively standard statistical analysis of DHS data: the key results of the DHS surveys are publicly available, and the dataset is freely available for researchers to analyse. It is surprising to me that such analysis has not already been done. I propose the authors to first run a systematic literature search to check what studies using this data have already been conducted. The main results are no surprise and in my view it is already common knowledge that factors such as high age or residing in an urban setting are associated with positive HIV status.

Response: Thank you for the suggestion to do a review. We conducted a rapid review and evidence mapping before the study started. A few studies already used DHS data for some aspects of our research, but those studies focused on individual countries while our own paper is regional and comparing progress over the years. We made references to some of those papers in the background and discussion sections.

2) The DHS data has a great amount of different variables. How did the authors choose the variables that were included in the analysis? Most variables were such that are already known to some extent correlate with HIV status, so the added value of this study seems limited. If the authors used a systematic approach to select the variables for the analyses (or e.g. followed the results of a previous analysis), this should be described in detail.

Response: The determinants were selected based on previous similar studies. We have made references to the papers. 

3) It is a bit unclear why the AIDSinfo prevalence data were included in this analysis. The analysis of time trends seems completely independent of the DHS data? Please note that the AIDSinfo estimates are themselves a result of models that use different data sources, so to me a simple description of such estimates does not count as original research.

Response: The reason for using both DHS and AIDSInfo data is for triangulation purposes. AIDSInfo data was also included to appreciate the prevalence trend over a period in the southern African region.

Detailed comments:

4) Abstract, Methods: Please be more specific of the actual analyses that you conducted (multivariable regression analysis of DHS data; autocorrelation analysis; etc).

Response: Thank you for the suggestion. We have added specific analyses. 

5) Introduction, last sentence: 73% suppression is essentially equivalent with the achievement of the 90-90-90 target, so no need to distinguish

Response: Thank you for the comments. 

6) Methods: Please be more clear that you used the DHS data and describe the data according to the usual DHS structure so that it's clear what datasets were used.

Response: Thank you for the suggestion. We have made the correction. 

7) The type of analysis seems not to be mentioned anywhere. Presumably it was logistic regression?

Response: It was logistic regression.

8) Be careful about the terms: an analysis adjusted for several variables is multivariable, not multivariate, and if you report crude odds ratios, these are results of univariable, not multivariable, analysis

Response: Thank you for the comments.

9) Under data analysis point d), it is unclear what you mean by “recommended”.

Response: The statement has been corrected.

10) Data analysis point e), how is the timing (“for each of the available data years”)? To my understanding only one (the latest) dataset from each country was analysed?

Response: Thank you. The statement has been corrected. 

11) Results: The paragraph “HIV prevalence trend…” is difficult to understand. The methods for estimating the yearly decrease are not given - did you use some formal statistical method, or is this simply the difference between 2020 and 2011 divided by 10 on the geometric scale? Since the direction in South Africa changed, how is it possible to give an annual increase until 2020? As suggested in the earlier comment, the added value of this analysis is unclear as it’s not related to the DHS data.

Response: Thank you for the comments. As we stated earlier, the inclusion of figure 1 was for triangulation purpose and as a supportive step to show how each country did over a decade since we could not access such long-term data for DHS for all the countries. South Africa had average annual increase unlike other countries that had reductions.

12) In my view, the most interesting results are those of the geospatial analysis. It would be nice to show these figures in the main manuscript.

Response: Thank you for the suggestions.

13) Discussion, second paragraph: It seems that prevalence and incidence are mixed up in this paragraph (and for example, it is unclear what “15% new infections in RSA in 2020” actually means). Please note that high prevalence is not necessarily a negative thing: with the expansion of ART coverage, people living with HIV are now living much longer and thus their share of the total population should not decrease too rapidly. The fact that HIV prevalence has continued to increase in RSA over time doesn’t necessarily imply that the epidemic is not under control (although this can of course be one reason). To my understanding the study only reported prevalence, not incidence, data, so nothing certain can be said about the actual epidemic control.

Response: Thank you for the comments. We have corrected the statement.

14) Moreover, I strongly disagree that reduction in HIV prevalence would be a result of religious practices. During the study period, all mentioned countries have moved from the limited ART coverage and CD4 dependent thresholds etc towards universal ART access to all PLHIV, and it is well known that successful ART minimized onward transmission. I would argue that the decrease in prevalence (and in particular incidence) is mainly attributable to high ART coverage, increased knowledge, less stigma etc.

Response: Thank you for the comment.

15) The language needs some (minor) editing.

Response: Thank you for the comment. We did a comprehensive editing.

---

## [Decision Letter · Decision Letter 1]

26 Sep 2023

PONE-D-22-32245R1Spatial distribution and determinants of HIV high burden in the Southern African sub-regionPLOS ONE

Dear Dr. Adetokunboh,

Thank you for submitting your manuscript to PLOS ONE. After careful consideration, we feel that it has merit but does not fully meet PLOS ONE’s publication criteria as it currently stands. Therefore, we invite you to submit a revised version of the manuscript that addresses the points raised during the review process. Thank you for revising your manuscript following the comments of the two previous reviewers. One of the previous reviewers is satisfied with the changed while the other was unavailable to provide comments this time. It was considered necessary to obtain the comments from an additional reviewer. The reviewers have raised remaining significant scientific concerns about the study that need to be addressed in a revision. 

Please revise the manuscript to address all the reviewer's comments in a point-by-point response in order to ensure it is meeting the journal's publication criteria. Please note that the revised manuscript will need to undergo further review, we thus cannot at this point anticipate the outcome of the evaluation process.

We look forward to receiving your revised manuscript.

Kind regards,

Miquel Vall-llosera Camps

Senior Editor

PLOS ONE

Reviewers' comments:

Reviewer's Responses to Questions

**Comments to the Author**

1. If the authors have adequately addressed your comments raised in a previous round of review and you feel that this manuscript is now acceptable for publication, you may indicate that here to bypass the “Comments to the Author” section, enter your conflict of interest statement in the “Confidential to Editor” section, and submit your "Accept" recommendation.

Reviewer #1: All comments have been addressed

Reviewer #3: (No Response)

2. Is the manuscript technically sound, and do the data support the conclusions?

Reviewer #1: Yes

Reviewer #3: Yes

3. Has the statistical analysis been performed appropriately and rigorously? 

Reviewer #1: Yes

Reviewer #3: Yes

4. Have the authors made all data underlying the findings in their manuscript fully available?

Reviewer #1: Yes

Reviewer #3: Yes

5. Is the manuscript presented in an intelligible fashion and written in standard English?

Reviewer #1: Yes

Reviewer #3: Yes

6. Review Comments to the Author

Reviewer #1: (No Response)

Reviewer #3: The authors stated, "Combining spatial analysis and the evaluation of the determinants of the HIV burden in Southern African countries is essential for better understanding of the disease dynamics in high burden settings." However, the authors failed to effectively persuade why this combination leads to a qualitative improvement in understanding the disease dynamics. While the authors explained the importance of spatial analysis of HIV infection and acknowledged the limited relevant previous studies in sub-Saharan Africa, this study only added two new countries on it, whereas four countries were already reported in Bulstra et al.'s study published in 2020. The significance of this addition is questionable.

Regarding the determinants of HIV, the unique contribution of this study remains unclear. The authors mentioned that "previous studies were either restricted to just a locality or conducted more than a decade ago." However, there are numerous studies available from other countries, raising concerns about the need to revisit the Southern African region for further research. The authors did not reference previous studies sufficiently, which weakens the justification for their current investigation.

To strengthen the study's rationale, it is crucial for the authors to thoroughly examine and reference previous reports. They must identify sufficient reasons for studying this subject in the Southern African region and clearly articulate the novel insights they aim to bring to the field.

There are some other comments:

1) This study utilized a secondary dataset; therefore, the authors explained the source of the original dataset and the process of extracting the data for their research. It is important to note that the authors did not personally select survey sites or conduct data collection. However, the statements presented in the methods section are somewhat confusing. To improve clarity, further elaboration is required to clearly distinguish the steps involved in obtaining the secondary dataset from the actual data collection procedures, which were not carried out by the authors themselves.

2) Table 1: The superscript symbol "*", was not explained elsewhere in the document. Please provide a clear explanation of its meaning to avoid any confusion for the readers.

3) Figure 1: To provide more detailed information, please mark the data points and connect them with lines instead of using continuous lines.

4) Table 3 not ‘Table 2’ shows the relationship between socio-demographic factors determinants.

5) S1 Table: This table contains the crude (univariate) results. Therefore, please remove 'multivariate' from the heading and state that the 'adjusted' results mean adjustment for all other predictive variables.

6) Please provide successive explanations for Figure S1-S6 to improve readability and help readers understand the content more easily.

7) Providing a clear image using hotspot cluster windows may offer advantages. However, it is important to note that the information presented is less detailed than that of a usual spatial mapping. To improve the clarity of the analysis, the authors should either provide a persuasive explanation of the advantages of using hotspot cluster windows or consider changing the approach to a usual mapping technique that offers more comprehensive spatial information.

8) “This finding contradicts Parkhurst's assertion that the relationship between HIV infection and household wealth index did not show reliable inclinations among Africans [22].” : Parkhurst (2010) stated that “the relationship between the prevalence of HIV infection and household wealth quintile did not show consistent trends in all countries. In particular, rates of HIV infection in higher-income countries did not increase with wealth.” The results of this study, where wealth level was found to be significant in all the countries except Zimbabewe, do not contradict Parkhurst’s report that the relationship is not consistent.

9) Conclusion needs revision to show the results and implication of this study.

7. PLOS authors have the option to publish the peer review history of their article (what does this mean?). If published, this will include your full peer review and any attached files.

Reviewer #1: No

Reviewer #3: **Yes: **Hae-Young Kim

---

## [Author Response · Author response to Decision Letter 1]

23 Nov 2023

REVIEWER’S COMMENTS AND RESPONSES

We would like to thank the reviewer for taking the time to provide valuable comments and suggestions. We believe that these insights have significantly improved the quality and presentation of our manuscript. We have addressed all the comments and made the necessary changes to the manuscript. Below, you will find our responses (highlighted in green) and the corresponding modifications to the manuscript (highlighted in yellow).

Reviewer #3: The authors stated, "Combining spatial analysis and the evaluation of the determinants of the HIV burden in Southern African countries is essential for better understanding of the disease dynamics in high burden settings." However, the authors failed to effectively persuade why this combination leads to a qualitative improvement in understanding the disease dynamics. While the authors explained the importance of spatial analysis of HIV infection and acknowledged the limited relevant previous studies in sub-Saharan Africa, this study only added two new countries on it, whereas four countries were already reported in Bulstra et al.'s study published in 2020. The significance of this addition is questionable.

Regarding the determinants of HIV, the unique contribution of this study remains unclear. The authors mentioned that "previous studies were either restricted to just a locality or conducted more than a decade ago." However, there are numerous studies available from other countries, raising concerns about the need to revisit the Southern African region for further research. The authors did not reference previous studies sufficiently, which weakens the justification for their current investigation.

To strengthen the study's rationale, it is crucial for the authors to thoroughly examine and reference previous reports. They must identify sufficient reasons for studying this subject in the Southern African region and clearly articulate the novel insights they aim to bring to the field.

Response: We thank the reviewer for the comments and suggestions, particularly for highlighting the need for a clearer justification. We have added several paragraphs to various sections of the paper to strengthen the report.

The progress in addressing the HIV epidemic is dynamic, and many sub-Saharan African countries have achieved notable advancements in reducing new infections and improving access to treatment and care; however, several Southern African countries are still battling relatively high HIV prevalence rates. This makes the area a region of interest that needs to be thoroughly investigated [2].

A geospatial analytical study characterizing areas with high levels of HIV transmission among seven countries in Eastern and Southern Africa regions shows that, among young adults, there were areas with relatively high prevalence alternating with low prevalence areas, signifying the presence of areas with high levels of HIV transmission [7]. However, this study is limited by the restriction of the participants to only young adults (15–29 years of age) and data that were collected as far back as 2008. Using data obtained from all adult age groups is expected to be more robust and informative than focusing on a single age group, as there are important differences in epidemiology, biology, and behavior across age groups [8]. A study on mapping HIV prevalence in sub-Saharan Africa shows significant local variations in the rate and direction of change in HIV prevalence between 2000 and 2017 in the region [9]. These findings demonstrate how significant local variations can be masked by national-level trends, especially when several important age cohorts are overlooked.

This work contributes to the existing body of knowledge by incorporating two fundamental approaches: geospatial analysis and multivariable logistic regression. By merging spatial pattern analysis with logistic regression, we achieve a more robust and reliable exploration of geographic, HIV, and sociodemographic data. This combined approach enhances precision in identifying hotspots and also pinpoints local drivers of disease dynamics. It leverages the individual strengths of each technique, fostering a more comprehensive understanding of the geospatial distribution of HIV in Southern Africa. These methods serve as a valuable framework in various domains, including public health, resource management, urban planning, and environmental science. It also provides a premise and reference point for future studies aimed at detecting the potential fading, emergence, or re-emergence of HIV hotspots. This, in turn, can guide timely, spatially targeted interventions. The outcomes will inform public health strategies, policy development, resource distribution, and the understanding of trends over time, all of which are crucial for combating the disease and enhancing the lives of those affected by HIV.

1) This study utilized a secondary dataset; therefore, the authors explained the source of the original dataset and the process of extracting the data for their research. It is important to note that the authors did not personally select survey sites or conduct data collection. However, the statements presented in the methods section are somewhat confusing. To improve clarity, further elaboration is required to clearly distinguish the steps involved in obtaining the secondary dataset from the actual data collection procedures, which were not carried out by the authors themselves.

Response: Thank you for your comment. We added statements on how the data were collected and analyzed.

The survey data for this study was a cross-sectional representative sample of households extracted from the Measure DHS website (http://www.dhs program.com) [9]. The sampling frame for the surveys consisted of enumeration areas across each included country. The surveys consisted of a two-stage sample design with standardized questionnaires administered to the participants [9]. Trained interviewers gave questionnaires to selected participants in each of the included countries during the data collection stage. Permission was granted for data extraction by Measure DHS and the data were downloaded in STATA format. 

DHS survey samples were selected using a stratified, two-stage cluster design with enumeration areas as the primary sampling unit and the households as the secondary sampling units in each of the participating countries. During the interview, individuals who met the eligibility requirements and were between the ages of 15 and 49 were encouraged to test for HIV voluntarily. After that, the skilled interviewers obtained dried blood spot samples from finger pricks for an HIV test. The anonymous linked procedure serves as the foundation for the sample collection and analysis protocol. The DHS program methodology allows for combining HIV test results with other information gathered based on each respondent's unique questionnaire. The DHS program was funded by various development partners, including the Global Fund to Fight AIDS, Tuberculosis and Malaria, the United States Agency for International Development, etc., and technical assistance from MEASURE DHS, ICF International, Calverton, Maryland, USA. The corresponding National Ministry of Health or Research/Statistical agencies implemented the DHS program.

2) Table 1: The superscript symbol "*", was not explained elsewhere in the document. Please provide a clear explanation of its meaning to avoid any confusion for the readers.

Response: Thank you for the suggestion. A footnote has been added.

* Number of participants tested positive for HIV and percentage in parenthesis.

3) Figure 1: To provide more detailed information, please mark the data points and connect them with lines instead of using continuous lines.

Response: Thank you for your suggestion. The figure has been deleted.

4) Table 3 not ‘Table 2’ shows the relationship between socio-demographic factors and determinants.

Response: Thank you for the comment., we have modified it accordingly. 

5) S1 Table: This table contains the crude (univariate) results. Therefore, please remove 'multivariate' from the heading and state that the 'adjusted' results mean adjustment for all other predictive variables.

Responses: Thank you for your suggestion. We have added more information about the adjusted results.

6) Please provide successive explanations for Figure S1-S6 to improve readability and help readers understand the content more easily.

Responses: Thank you for your suggestion. We have provided additional explanations. See below. 

The study shows a Low-High HIV occurrence and cluster around the Central Region of Malawi; this implies that the area has low HIV prevalence, but its neighboring clusters have high HIV prevalence; the High-Low cluster in Zambezia Province - the area has a high HIV prevalence, but its neighboring areas have low values for the same variable; and the High-High cluster in Maputo Province of Mozambique - the area has a high HIV prevalence, likewise the neighboring clusters. In addition, there were also Low-Low clusters in the Omaheke, Khomas and Hardap Regions of Namibia – the clusters and neighboring clusters has low HIV prevalence; High-High cluster in Southern Province of Zambia; and Low-High cluster in Bulawayo Province and High-Low cluster in Mashonaland East Province of Zimbabwe.

7) Providing a clear image using hotspot cluster windows may offer advantages. However, it is important to note that the information presented is less detailed than that of a usual spatial mapping. To improve the clarity of the analysis, the authors should either provide a persuasive explanation of the advantages of using hotspot cluster windows or consider changing the approach to a usual mapping technique that offers more comprehensive spatial information.

Response: Thank you for your comment. We added statements on hotspot cluster windows.

We used kernel density estimation (KDE) because it can effectively detect clusters of events within a given geographic area. KDE is a hotspot mapping technique that produces a smooth and continuous surface map that illustrates gradients of the variation in event intensity across the study fields without being constrained by theme borders [42]. Its striking visual appeal makes it a popular mapping technique and one of the frequently used approaches for point pattern analysis [42,43]. KDE differs from other mapping techniques in that it uses a weighting function based on a constant bandwidth or search radius to generate a surface based on a nonparametric estimate of the intensity function across cell grids [44]. We can obtain the bandwidth and then determine the smallest distance at which clustering is most intense and significant [42].

8) “This finding contradicts Parkhurst's assertion that the relationship between HIV infection and household wealth index did not show reliable inclinations among Africans [22].” : Parkhurst (2010) stated that “the relationship between the prevalence of HIV infection and household wealth quintile did not show consistent trends in all countries. In particular, rates of HIV infection in higher-income countries did not increase with wealth.” The results of this study, where wealth level was found to be significant in all the countries except Zimbabewe, do not contradict Parkhurst’s report that the relationship is not consistent.

Response: Thank you for your comment. We have rectified the statement.

HIV infection is assumed to be prevalent among the poor and with a lesser number among the middle and upper social class [26]. Gaumer et al. show that wealth effects on HIV prevalence are usually smaller and statistically insignificant; however, a high wealth index was not associated with higher HIV prevalence, while low wealth was associated with a higher risk among certain sections of the population. [27]. 

9) Conclusion needs revision to show the results and implication of this study.

Response: Thank you for your comment. We have added the following paragraph to the manuscript. 

Our findings highlight the significance of regional differences in HIV prevalence as well as the impact of sociodemographic determinants and sexual behaviors. Our results suggest that analysis of subnational data could provide reasonable estimates of the wide-ranging spatial structure of the HIV epidemic in selected Southern African settings. However, similar analyses should be conducted at district and municipality levels to assess community-level patterns. The spatial distribution of high burden areas for HIV in the selected countries was more pronounced in the major cities. Other measures such as prevention and identification of STIs should also be prioritized. Determinants such as individuals who are divorced or widowed, middle-aged women, and people who recently treated STIs, should be the focus of HIV prevention and control interventions in the Southern Africa Region. The findings of this study are useful in developing HIV prevention programs as it identifies the areas and communities that require additional resources and attention. Healthcare policymakers especially in resource-constrained communities in the Southern African region can apply our findings to identify areas to target for the design of interventions to reduce HIV transmission and inform the prevention and control programs. Lastly, HIV control interventions such as HIV/AIDS awareness campaigns, ART services and adherence support for those on ART should be focused on locations identified as hotspot clusters.

---

## [Decision Letter · Decision Letter 2]

19 Dec 2023

PONE-D-22-32245R2Spatial distribution and determinants of HIV high burden in the Southern African sub-regionPLOS ONE

Dear Dr. Adetokunboh,

Thank you for submitting your manuscript to PLOS ONE. After careful consideration, we feel that it has merit but does not fully meet PLOS ONE’s publication criteria as it currently stands. Therefore, we invite you to submit a revised version of the manuscript that addresses the points raised during the review process.

We look forward to receiving your revised manuscript.

Kind regards,

Jianhong Zhou

Staff Editor

PLOS ONE

Journal Requirements:

Reviewers' comments:

Reviewer's Responses to Questions

**Comments to the Author**

1. If the authors have adequately addressed your comments raised in a previous round of review and you feel that this manuscript is now acceptable for publication, you may indicate that here to bypass the “Comments to the Author” section, enter your conflict of interest statement in the “Confidential to Editor” section, and submit your "Accept" recommendation.

Reviewer #3: All comments have been addressed

2. Is the manuscript technically sound, and do the data support the conclusions?

Reviewer #3: Yes

3. Has the statistical analysis been performed appropriately and rigorously? 

Reviewer #3: Yes

4. Have the authors made all data underlying the findings in their manuscript fully available?

Reviewer #3: Yes

5. Is the manuscript presented in an intelligible fashion and written in standard English?

Reviewer #3: Yes

6. Review Comments to the Author

Reviewer #3: The revision seems improved. One recommendation was added.

On the Figure 1~Figure 6, please add the year that the data were obtained and rearrange them into from Figure 1 a) to Figure 1 g).

7. PLOS authors have the option to publish the peer review history of their article (what does this mean?). If published, this will include your full peer review and any attached files.

Reviewer #3: No

---

## [Author Response · Author response to Decision Letter 2]

23 Feb 2024

REVIEWER’S COMMENTS AND RESPONSES - REVISION 3

We would like to thank the reviewers for taking the time to provide valuable comments and suggestions. We believe that these insights have significantly improved the quality and presentation of our manuscript. We have addressed all the comments and made the necessary changes to the manuscript. 

1. Please remove your figures/ from within your manuscript file, leaving only the individual TIFF/EPS image files. These will be automatically included in the reviewer’s PDF.

Response: We thank the reviewer for the comments and suggestions. We have removed the figures from the manuscript as suggested. 

2. We note that Figures 1a-1f in your submission contain [map/satellite] images which may be copyrighted. All PLOS content is published under the Creative Commons Attribution License (CC BY 4.0), which means that the manuscript, images, and Supporting Information files will be freely available online, and any third party is permitted to access, download, copy, distribute, and use these materials in any way, even commercially, with proper attribution. For these reasons, we cannot publish previously copyrighted maps or satellite images created using proprietary data, such as Google software (Google Maps, Street View, and Earth). For more information, see our copyright guidelines: http://journals.plos.org/plosone/s/licenses-and-copyright.

1. You may seek permission from the original copyright holder of Figures 1a-1f to publish the content specifically under the CC BY 4.0 license.

Natural Earth (public domain): http://www.naturalearthdata.com/"

Response: We thank the reviewer for the suggestions. We were unable to get permission from the Measure DHS for the use of the GIS shapefiles and copyrighted maps. However, we used GADM database of Global Administrative Areas v 4.1.

GADM data are freely available for academic use and other non-commercial uses. Using the data to create maps for publishing academic research articles is allowed. Thus, could use the maps made with GADM data for figures in articles published by PLoS, Springer Nature, Elsevier, MDPI, etc. You are allowed (but not required) to publish these articles (and the maps they contain) under an open license such as CC-BY as is the case with PLoS journals and maybe the case with other open access articles.

3. Thank you for uploading your study's underlying data set. Unfortunately, the repository you have noted in your Data Availability statement does not qualify as an acceptable data repository according to PLOS's standards.

Response: Thank you for the suggestions. We have uploaded the data set on figshare: https://figshare.com/s/33e95ee4594a7c146e3b

REVIEWER’S COMMENTS AND RESPONSES - REVISION 2

We would like to thank the reviewer for taking the time to provide valuable comments and suggestions. We believe that these insights have significantly improved the quality and presentation of our manuscript. We have addressed all the comments and made the necessary changes to the manuscript. 

Reviewer #3: The revision seems improved. One recommendation was added.

On the Figure 1~Figure 6, please add the year that the data were obtained and rearrange them into from Figure 1A) to Figure 1F).

Response: We thank the reviewer for the comments and suggestions. We have renamed the figures as suggested.

---

## [Editor Report · Decision Letter 3]

24 Mar 2024

Spatial distribution and determinants of HIV high burden in the Southern African sub-region

PONE-D-22-32245R3

Dear Dr. Adetokunboh,

We’re pleased to inform you that your manuscript has been judged scientifically suitable for publication and will be formally accepted for publication once it meets all outstanding technical requirements.

Kind regards,

Jianhong Zhou

Staff Editor

PLOS ONE